# Unsupervised classification of brain-wide axons reveals the presubiculum neuronal projection blueprint

Diek W. Wheeler [1] ✉, Shaina Banduri[1], Sruthi Sankararaman[1], Samhita Vinay[1] & Giorgio A. Ascoli [1] ✉

We present a quantitative strategy to identify all projection neuron types from a given region with statistically different patterns of anatomical targeting. We first validate the technique with mouse primary motor cortex layer 6 data, yielding two clusters consistent with cortico-thalamic and intra-telencephalic neurons. We next analyze the presubiculum, a less-explored region, identifying five classes of projecting neurons with unique patterns of divergence, convergence, and specificity. We report several findings: individual classes target multiple subregions along defined functions; all hypothalamic regions are exclusively targeted by the same class also invading midbrain and agranular retrosplenial cortex; Cornu Ammonis receives input from a single class of presubicular axons also projecting to granular retrosplenial cortex; path distances from the presubiculum to the same targets differ significantly between classes, as do the path distances to distinct targets within most classes; the identified classes have highly non-uniform abundances; and presubicular somata are topographically segregated among classes. This study thus demonstrates that statistically distinct projections shed light on the functional organization of their circuit.

The classification of neurons in the mammalian nervous system has long been a focus of intensive investigation. While local features from slice preparations in vitro may suffice to infer the circuit roles of GABAergic interneurons[1–3], long-range projecting axons are crucial architectural elements of neural organization[4,5] constituting the conceptual and physical nexus between brain-wide circuits and synaptic communication[6]. Thus, projection axons have long been digitally traced from serial sections after in vivo labeling and light microscopy imaging[7–10]. At the same time, their macroscopic extent (~1 cm span; ~1 m cable length) and microscopic caliber (~100 nm branch thickness) combine into a formidable technological challenge for large-scale collection[11,12]. As a result, the number of completely reconstructed projection axons in any mammalian neural system has until recently remained into the low tens.

A source brain region projecting to N targets (where N typically ranges between 10 and 50 in the mouse cortex) could contain any combination of $2^N-1$ distinct axonal projection types. Such a combinatorics challenge requires a large-scale data collection for proper classification. Projects based on fluorescent Micro-Optical Sectioning Tomography (fMOST) technology[13–15] or the Janelia MouseLight platform[16], launched in recent years to address this need, produced nearly 10,000 mouse whole-brain single neuron reconstructions registered to a 3D Common Coordinate Framework (CCF)[17] with consensus anatomical labeling[18]. However, these newly available data do not themselves generate novel scientific insights, explain brain circuitry, or even disprove that axons might simply invade a random subset of the regional target areas[19]. Rigorous methods are needed to test the hypothesis that specific projection types exist, to characterize their identities, and to quantify their population sizes[20].

[1]Center for Neural Informatics, Krasnow Institute for Advanced Studies and Bioengineering Department, College of Engineering & Computing, George Mason University, Fairfax, VA, USA. ✉e-mail: dwheele5@gmu.edu; ascoli@gmu.edu

This study introduces an original technique to objectively identify projection-based neuronal classes. To ascertain whether a collection of axonal projections might result from essentially random variation within the constraints of regional connectivity or likely reflects distinct neuron types, we begin from the foundational criterion for classification: if a set of items belongs to segregated classes, their pairwise interindividual differences must be on average larger between than within classes. In other words, two items from the same class should tend to be more similar to each other than two items from separate classes. To implement this logic into a classification framework, we couple rigorous statistical testing with unsupervised hierarchical clustering. A unique strength of this approach is its entirely data-driven granularity: the continuous accumulation of new tracings will progressively refine the classification details with increasing statistical power. We can then characterize the identified projection classes by quantifying their population size, topographic soma distributions, and convergence and divergence patterns.

In the remainder of this article, we first propose a formal definition of and a quantitative solution for the classification problem. We validate our approach by applying it to layer 6 of the primary motor cortex, and then utilize it to study the presubiculum, a rather underinvestigated region of the mouse brain. We next quantify the neuronal population sizes of the presubicular projection classes and characterize the spatial distribution of their somata. Finally, we analyze the patterns of divergence and convergence of presubicular projection classes. We conclude by discussing the biological interpretations of these results.

## Results

### Quantitative solution of the classification problem

The axonal projections of each neuron in a source region can be represented as k-dimensional vectors, where k is the number of target

regions invaded by the source region. Each of the k components of the vector quantifies the number of axonal points within the corresponding region (Fig. 1; see "Choice of metric to quantify axonal extent" in "Methods"). We explore the null hypothesis, $H_0$, that all neurons from a source region belong to a single projection class (Fig. 2a), as opposed to the alternative hypothesis, $H_A$, that distinct projection classes exist from that source region (Fig. 2b). If two hypothetical classes exist, the projections will be more similar between neurons within a class and more different across classes (Fig. 2c). In such a two-class scenario, the combined within- and across-class distances would thus form a wider distribution than the distribution generated if all neurons belong to just a single class (Fig. 2d). To formally test $H_A$, we measure all pairwise differences between neurons (as arccosine vector distances, see "Methods"). We then generate the distribution of distances for $H_0$ by randomizing the projection patterns while preserving total axonal extent both by neuron and target region. We achieve this single-class continuum by iterative stochastic swapping of axonal points between neurons across two target regions (see Fig. 2e and "Methods"). We can then apply Levene's one-tail statistical test to ascertain whether the original distribution of pairwise distances has significantly larger variance than the randomized distribution. If the answer is positive, we must discard $H_0$ and accept $H_A$. Starting from the top node in an unsupervised hierarchical clustering tree, we can thus repeat Levene's test on the neurons of each of the two subtrees, continuing the process until none of the variance differences are statistically significant (Fig. 2f). When Levene's test fails (i.e., it provides a negative answer), the precise cutoff is determined independently of the other points of failure. Therefore, all neurons within a cluster (i.e., under the same Levene failure point) are statistically equivalent with respect to the axonal projection patterns across the target regions, but each cluster is independent of the other clusters. Moreover, there is no correspondence between the cutoff levels and the resulting number of neurons in each cluster.

### Validation of the approach

To validate the above research design, we first analyzed 52 MouseLight layer 6 neurons from the primary motor cortex[21] (Source data are provided as a Source Data file). This anatomical area is known to contain two distinct projection classes with well-defined subdivisions: cortico-thalamic (CT) and cortico-cortical or intra-telencephalic (IT) neurons[22]. The variance of the distribution of pairwise axonal projection differences of these neurons was significantly larger than that of the randomized projections ($p = 6.46 \times 10^{-51}$; variance of real data = 373.4; variance of randomized data = 195.7), indicating the existence of distinct clusters. However, both subtrees after the first split of unsupervised hierarchical clustering returned a non-significant Levene's test (IT: $p = $ N/A; variance of real data = 219.9; variance of randomized data = 240.0; CT: $p = 0.24$; variance of real data = 295.1; variance of randomized data = 264.0), revealing exactly two clusters (Fig. 3a). The first cluster, consisting of 21 neurons, projected almost exclusively to motor cortical targets; the second cluster of 31 neurons projected primarily to thalamic targets (Fig. 3b–d). These patterns were fully consistent with the axonal pathways of the IT and CT neuronal classes, respectively. This finding thus corroborates the validity of employing Levene's test of variance on pairwise difference distributions to identify statistically distinct classes in unsupervised hierarchical clustering.

### Classification of projection neurons from mouse presubiculum

We then applied our analytic technique to a lesser-explored source region of the mouse brain: the presubiculum. Unsupervised clustering and the test of variance demonstrated that the 93 MouseLight neurons from the presubiculum form five distinct projection classes (Fig. 4a–c). We designate each class by a letter (A-E) followed by the number of neurons in the class (Fig. 4c). The first class, A38, primarily targets the

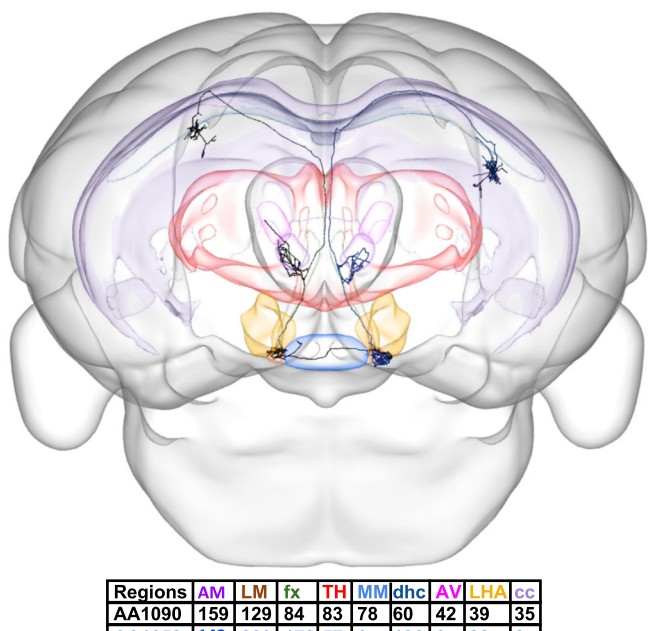

| Regions | AM | LM | fx | TH | MM | dhc | AV | LHA | cc |
|---|---|---|---|---|---|---|---|---|---|
| AA1090 | 159 | 129 | 84 | 83 | 78 | 60 | 42 | 39 | 35 |
| AA1058 | 143 | 209 | 178 | 57 | 0 | 136 | 0 | 38 | 2 |

**Fig. 1 | Brain-wide neuronal projections.** CCF-registered reconstruction of two presubicular neurons (brain depiction and neurons AA1090 in black and AA1058 in blue from the Janelia MouseLight project) invading 9 regions out of 40 potential targets along with the numbers of axonal points of the neurons in each highlighted region (posterior view of brain). Source data are provided as a Source Data file. CCF common coordinate framework, AM anteromedial nucleus, AV anteroventral nucleus, cc corpus callosum, dhc dorsal hippocampal commissure, fx fornix, LHA lateral hypothalamic area, LM lateral mammillary nucleus, MM medial mammillary nucleus, TH other thalamic nuclei.

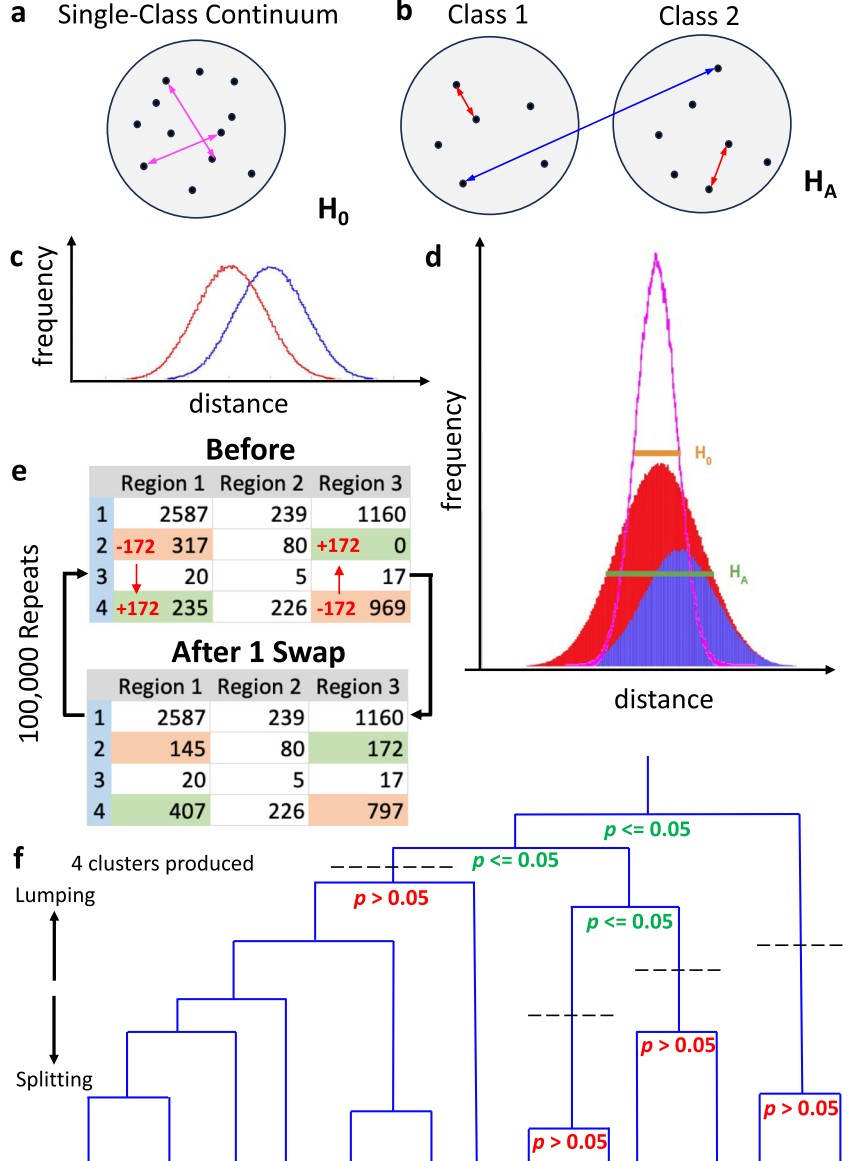

**Fig. 2 | Definitions of neuron classes and clustering methods. a** In a single-class scenario, the distribution of differences between neurons can be calculated for all neuron pairs (pink double-arrows). **b** If two distinct classes exist, neurons (represented here as black dots) will tend to have more similar projections within their class (red double-arrows) and more different ones across classes (blue double arrow). **c** The differences within the classes (red distribution) will be smaller than those between classes (blue distribution). **d** The distribution of the combined frequency of differences, in a multi-class scenario (red-blue stacked areas; green half-height width), will be wider than that of a single-class distribution (pink curve; orange half-height width). **e** Diagram showing the randomization of projection patterns through the repeated pairwise swapping of axonal point counts between two neurons across two of their potential target regions, which preserves the column (for a given region) and row (for a given neuron) sums of the matrix. This swapping results in a projection pattern continuum that matches with the overall distribution representing the 1-class null hypothesis. **f** Unsupervised hierarchical clustering groups a set of neurons into classes based on their relative pairwise differences or similarities, as modeled by a binary dendrogram. The top (root) of the dendrogram represents all neurons lumped into the same class, while the bottom (leaves) shows every neurons split into separate classes. The $p$ value that determines whether to keep splitting is derived from Levene's test based on a one-way ANOVA of the absolute data values and the group means to which the data values belong.

lateral entorhinal cortex (LEC), accounting for 82% of axonal extent outside of the presubiculum. This class also invades the dorsoventral (granular) retrosplenial cortex as well as the hippocampal formation (dentate gyrus, CA3, CA2, CA1, and subiculum). The second class, B27, mainly targets the dorsal portion of the medial entorhinal cortex (dMEC), accounting for 92.5% of extra-presubicular axonal extent, as well as retrohippocampal zone and parasubiculum. Class C3 neurons mostly target the contralateral dMEC (42%) and LEC (40%), subiculum (14%), and parasubiculum (4%) through extensive callosal and commissural fibers. Class D19 has the most complex (and unreported) pattern of innervation: in addition to major projections to the subiculum (40.8%) and dentate gyrus (16.3%), it is the sole source of projections to the lateral (agranular) retrosplenial cortex, to the hypothalamus (including the lateral mammillary nucleus and 18 additional nuclei), and to the superior and inferior colliculi in the midbrain. This neuronal class also projects to a subset of 8 thalamic nuclei, including the medial part of the anterior thalamic nucleus (ATN) and the lateral geniculate nucleus. Lastly, class E6 projects to a complementary set of 14 other thalamic nuclei, including the ventral, dorsal, anterior, and lateral parts of the ATN and the medial geniculate nucleus. Neurons from all five projection classes also have substantial collaterals within the presubiculum. Examples of projection neurons

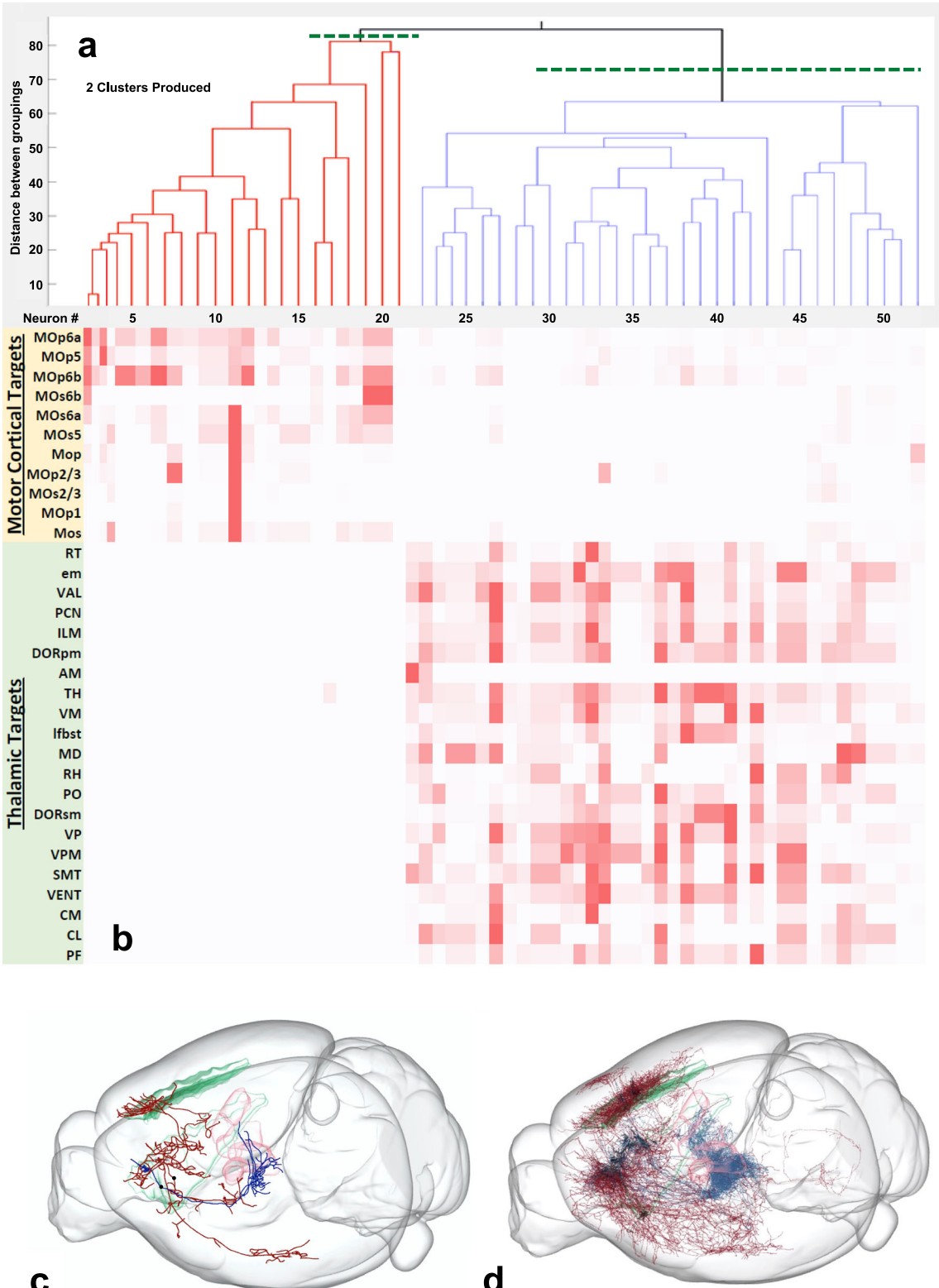

**Fig. 3 | Primary motor cortex L6 (IT vs. CT). a** Representation of the two clusters produced by Levene's one-tailed test for the equality of variances and unsupervised hierarchical clustering, using MouseLight neurons from the primary motor cortex, layer 6 (*n* = 52). **b** Colormap of the axonal distributions of neurons (columns) across anatomical regions (rows), with darker shades representing more axonal projections. The axonal points for the thalamic targets are more numerous than those for the motor cortical targets by a factor of two. Source data are provided as a Source Data file. **c** CCF-registered reconstructions of the axonal pathways of representative

IT (intra-telencephalic, red, AA0876) and CT (corticothalamic, blue, AA0398) neurons with semitransparent surfaces of primary motor cortex layer 6 (green) and selected thalamic nuclei (pink). The two black dots indicate the cell body locations of the two representative cells from each class. **d** CCF-registered reconstructions of the axonal pathways of all IT and CT neurons (see Source Data for a full listing of the 52 neurons depicted) in the MouseLight sample (same color coding). The neurons and brain depictions in panels (**c**) and (**d**) are from the Janelia MouseLight project.

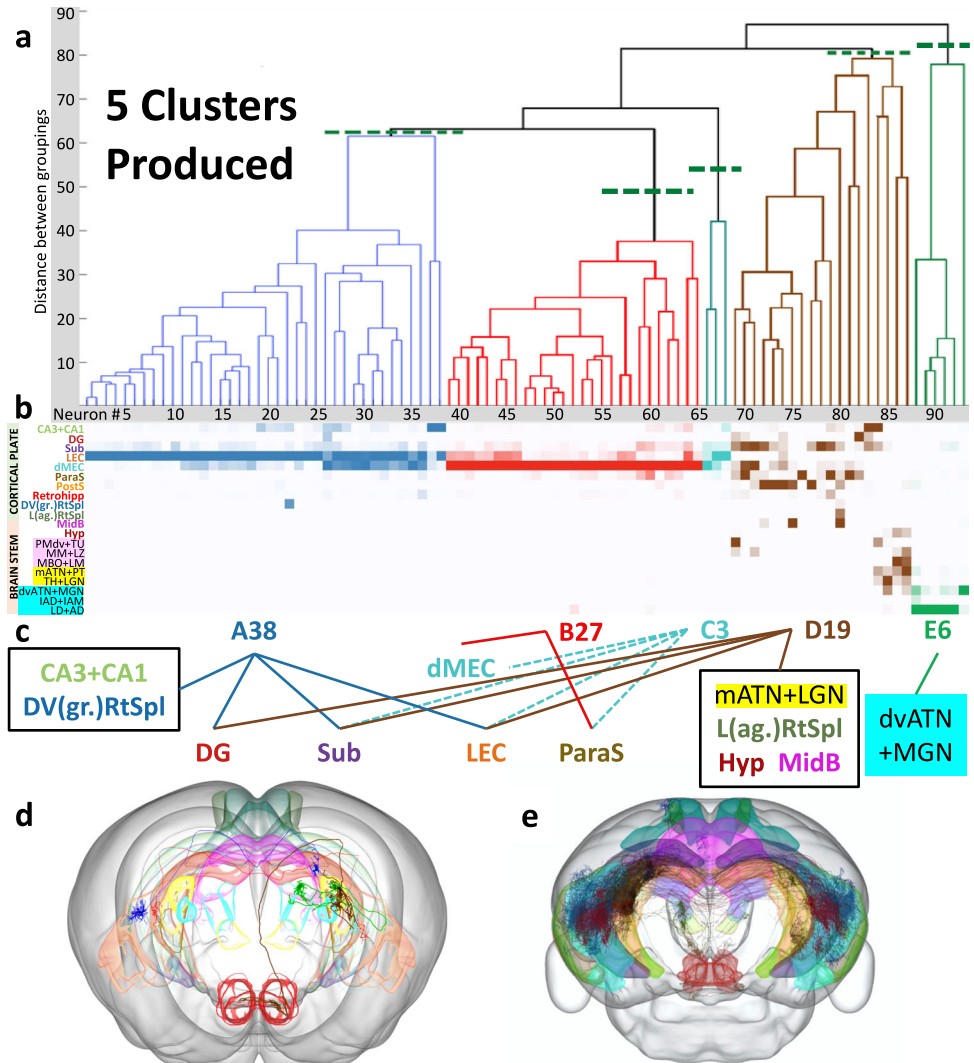

**Fig. 4 | Classification of projection neuron types in the presubiculum.**
**a** Representation of 5 axonal clusters produced by Levene's test and unsupervised hierarchical clustering of neurons from the presubiculum (*n* = 93). **b** Colormap of the axonal distributions of neurons (columns) across anatomical regions (rows), with darker shades representing more axonal projections. Parcel names highlighted in pink are hypothalamus related. Parcel names highlighted in yellow and light blue are thalamus related. Source data are provided as a Source Data file. **c** Neuron-to-target assignments for the identified axonal projection classes and corresponding anatomical regions (dotted line: contralateral). **d** Anterior view of the mouse brain with a CCF-registered reconstruction of one neuron from each class (cluster A, blue, AA0021; cluster B, red, AA0724; cluster C, cyan, AA0168; cluster D, brown, AA0031; cluster E, green, AA0244). Color coding of neurons and semitransparent anatomical areas shown in (**a**, **b**, and **c**). **e** Posterior view of the brain with CCF-registered reconstructions of all 93 MouseLight presubicular neurons (see Source Data for a full listing of the neurons depicted). The highlighted

parcels are the same as those depicted in panel (**c**), with the same color coding. The neurons and brain depictions in panels (**d**) and (**e**) are from the Janelia MouseLight project. CA3 + CA1 Cornu Ammonis areas 3 and 1, DG dentate gyrus, Sub subiculum, LEC lateral entorhinal cortex, dMEC dorsal portion of the medial entorhinal cortex, ParaS parasubiculum, PostS postsubiculum, Retrohipp retrohippocampal region, DV(gr.)RtSpl dorsal and ventral (granular) retrosplenial cortex, L(ag.)RtSpl lateral (agranular) retrosplenial cortex, MidB midbrain, Hyp hypothalamus, PMdv + TU dorsal and ventral premammillary nucleus and tuberal nucleus, MM + LZ: medial mammillary nucleus and hypothalamic lateral zone, MBO + LM mammillary body and lateral mammillary nucleus, mATN + PT medial anterior thalamic nucleus and parataenial nucleus, TH + LGN thalamus and lateral geniculate nucleus, dvATN + MGN dorsal and ventral anterior thalamic nucleus and medial geniculate nucleus, IAD + IAM interanterodorsal and interanteromedial nucleus of the thalamus, LD + AD lateral dorsal and anterodorsal nucleus of thalamus.

from each of the presubicular projection classes are depicted in Fig. 4d–e.

**Presubicular classes have non-uniform population sizes**
Next, we quantified the proportion of neurons in the mouse presubiculum that belong to each projection class. To this aim, we extracted the anterograde tract tracing density distributions from the Allen Institute regional connectivity atlas and matched the fractions of neurons in every class based on their axonal patterns by numerical optimization (see Non-Negative Least Squares in Methods; Source data are provided as a Source Data file). The results converged with very

small residual error (<0.0006%) indicating a near-exact correspondence between single-neuron and regional projections. Fully sampling neurons from across the presubiculum, Class D19, reaching the midbrain, hypothalamus, lateral (agranular) retrosplenial, and the lateral geniculate (visual thalamus) accounted for the greatest portion (38.1%) of neurons. Class A38, targeting the hippocampus, subiculum, dorsoventral (granular) retrosplenial cortex, and lateral entorhinal cortex (what pathway), accounted for the second largest share (30.6%) of neurons. Class B27, projecting to the parasubiculum and medial entorhinal cortex (where pathway) consisted of 16.3% of projection neurons. Class E6, focused on other thalamic nuclei including medial

geniculate (auditory), was responsible for 13.7% of presubicular neurons. The diffuse contralateral projections of class C3 comprised the remaining 1.3%.

When accounting for these relative proportions together with the MouseLight axonal projections, we can estimate the contribution of each class to the presubicular projections in each collection of target regions. In particular, the dentate gyrus receives 21% of its presubicular afferents from class A38 and 79% from class D19. The subiculum receives 69% of its presubicular afferents from class D19, 30% from class A38, and 1% from class C3. The lateral entorhinal cortex receives 99% of presubicular afferents from class A38 and 1% from class C3. The dorsal medial entorhinal cortex and parasubiculum receive 99% of presubicular afferents from class B27 and 1% from class C3. All other regions are targeted by individual classes: CA3, CA1, and the dorsoventral (granular) retrosplenial cortex by A38; the midbrain, hypothalamus, lateral (agranular) retrosplenial cortex, and part of the thalamic nuclei including medial ATN and lateral geniculate nucleus by D19; and the rest of the thalamic nuclei including dorsoventral ATN and medial geniculate nucleus by E6.

### Somata distribution reveals class topographic organization

Computational geometry analysis of soma locations within the presubiculum demonstrated a clear spatial separation among the four main projection classes: A38, B27, D19, and E6 (the smallest class, C3, is largely contralateral projecting). Specifically, the convex hull volume of each neuron class overlapped only minimally (~5–20%) with that of other neuron classes (Fig. 5a–c). In particular, class A38 was positioned more rostrally and dorsally relative to the caudal-ventral position of class B27, with approximately 14% of overlap (Fig. 5a). The overlap of A38 was maximal with D19 (21%); however, while most A38 neurons had a selective somatic concentration in layer 2 (34/38: 89.5%), D19 had a somatic distribution across all 3 presubicular layers: 21% in layer 1 and 26% in layer 3 (Fig. 5b). Class E6 had the most lateral positioning resulting in almost complete segregation from the other projection classes: there were so few overlapping somata that a proper convex hull volume of the overlap could not be calculated (Fig. 5c, d).

### Efferent path distances from the same neurons vary by target

We tested whether the path distances from presubicular neurons of a given projection class differed across their divergent target regions (Fig. 6). In these analyses of divergence, ipsilateral and contralateral targets were considered separately, as the latter are systematically farther than the former. For class A38 neurons, projection distances to the ipsilateral lateral entorhinal cortex, subiculum, and dentate gyrus are significantly shorter than those to the ipsilateral hippocampus; moreover, projection distances to the ipsilateral lateral entorhinal cortex are significantly longer than those to the ipsilateral subiculum and dentate gyrus. Similarly, projection distances to the contralateral subiculum and lateral entorhinal cortex are significantly shorter than those to the contralateral hippocampus. Thus, presubicular efferent path distances differ less between ipsilateral and contralateral hippocampus than between other targets across brain hemispheres (Fig. 6a). For class B27, projections to the ipsilateral parasubiculum have significantly shorter paths than those to medial entorhinal cortex, dorsal zone, but the distances are comparable in the contralateral case (Fig. 6b). Finally, for class D19, projections both to the ipsilateral medial anterior thalamic nucleus and lateral geniculate nucleus, and to the ipsilateral hypothalamus and lateral mammillary nucleus combined have significantly longer paths than those to the ipsilateral midbrain (Fig. 6c).

### Afferent path distances to the same region vary by class

Next, we asked whether the axons from neurons of distinct projection classes converging onto their shared targets had different path distances. With the sole exception of the dentate gyrus, all target regions displayed a significant dependence of path distance on the presubicular neuron class (Fig. 7). For the ipsilateral medial entorhinal cortex, dorsal zone, projections from E6 and D19 have shorter distances than those from B27 and A38, and projections from B27 have significantly shorter distances than those from A38. For the contralateral medial entorhinal cortex, in contrast, projections from B27 have significantly longer distances than those from A38 (Fig. 7a). For the ipsilateral parasubiculum, path distances from D19 are significantly longer than those from B27 (Fig. 7b). Finally, for the contralateral subiculum, parasubiculum, and lateral entorhinal cortex, path distances from B27 are significantly longer than those from A38 (Fig. 7b–d).

## Discussion

This study introduced an original method to objectively identify projection-based neuronal classes by pairing the Levene's test with unsupervised hierarchical clustering. We first conducted a confirmatory study on layer 6 of the primary motor cortex to verify that the proposed technique could reproduce known projection types in a previously explored area of the mammalian brain. The results yielded two clusters with axonal projections consistent with those of the corticothalamic and intratelencephalic neuron classes found in past studies, thereby confirming the validity of the technique[23].

Levene's test was chosen because it is not dependent on the data distributions being normal. Given the size of current available data, normality cannot be assured. As the accumulation of data increases by several orders of magnitude, it is possible that other statistical tests, such as an *F*-test, could be used instead. Another form of unsupervised clustering, such as K-means, could be utilized to achieve similar ends to what we were able to achieve. The important aspect is that the method be unsupervised, such that the data themselves direct the clustering without any user input.

To test whether the technique could lead to novel insights, we then applied it to the presubiculum, a region with crucial cognitive function[24], yet few studies on its circuitry[25]. The results yielded five clusters, indicating distinct neuron classes, which led us to reject the null hypothesis that projection neurons exhibit random variation within the constraints of regional connectivity from the presubiculum. In an earlier study[26], retrograde tracing identified five classes of neurons projecting from the presubiculum, which target the retrosplenial cortex (corresponding to our class A38), contralateral subiculum (class C3), medial entorhinal cortex (class B27), anterior thalamic nucleus (class E6), and lateral mammillary nucleus (class D19). Our results confirm the existence of these five classes and add new information that reveals patterns of divergence (e.g., class A38 projects to the retrosplenial cortex, dentate gyrus, subiculum, and entorhinal cortex), convergence (e.g., the subiculum receives projections from classes A38, contralateral C3, and D19), and specificity (e.g., class E6 projects exclusively to the medial geniculate nucleus, and all hypothalamic regions receive projections solely from class D19; see summary Fig. 8).

The proposed clustering technique correctly distinguishes cortical (classes A38, B27, and C3) from subcortical (D19 and E6) pathways in the second binary split in the hierarchical classification. These results also add cellular level details to previously reported presubicular projections to retrosplenial cortex and thalamic reticular nuclei[27], as well as a broader circuit context to the characterization of individual presubicular neurons targeting the medial entorhinal cortex[28].

Furthermore, our findings reveal that several target regions are spatially subdivided according to the differing inputs between classes. These regions include the entorhinal cortex (lateral projections mainly from class A38 and medial projections primarily from class B27), retrosplenial cortex (dorsoventral granular projections almost exclusively from class A38 and lateral agranular projections solely from class

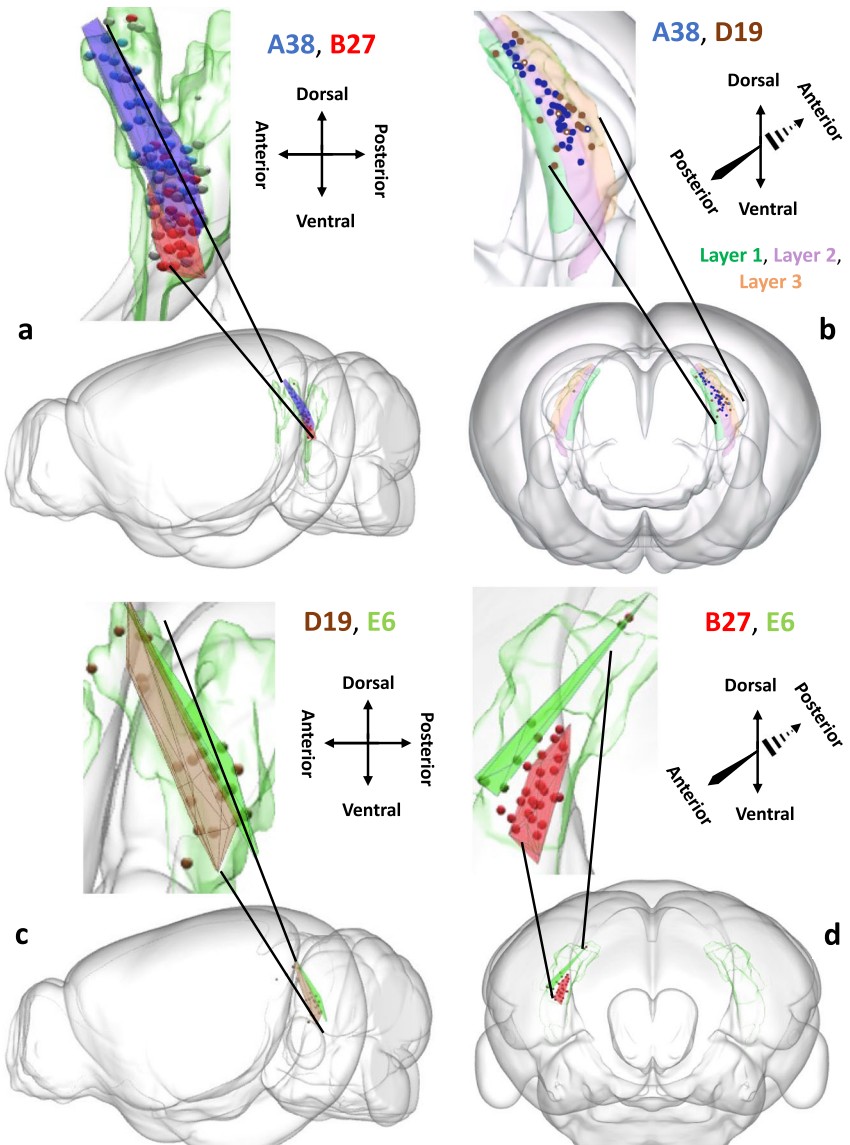

**Fig. 5 | Spatial distributions of somata in the presubiculum across projection classes.** Convex hulls of neurons (spheres) from classes A38 (blue), B27 (red), D19 (brown), and E6 (green), and semitransparent presubiculum (green). **a** Left sagittal view of A38 and B27. **b** Layer 1 (green), layer 2 (purple), and layer 3 (orange) of the presubiculum are highlighted in an anterior coronal view, with somata from A38 in blue and D19 in brown. Most of the A38 somata are concentrated in layer 2, while the D19 somata tend to be more concentrated in layers 1 and 3. Somata that do not follow this pattern are indicated with a white dot inside of the circle. **c** Left sagittal view of D19 and E6. **d** Posterior coronal view of B27 and E6. The brain depictions in all panels are from the Janelia MouseLight project, and the spheres were generated with MATLAB.

D19), and thalamus (medial anterior thalamic nucleus and lateral geniculate nucleus projections principally from class D19 and dorsoventral anterior thalamic nucleus and medial geniculate nucleus projections predominantly from class E6). Some of these regional subdivisions also have known functional distinctions: for instance, the medial entorhinal cortex specializes in spatial representation while the lateral entorhinal cortex specializes in integrating sensory input[29]. Among the thalamic geniculate nuclei, the medial geniculate nucleus is part of the auditory pathway, whereas the lateral geniculate nucleus is part of the visual pathway[4].

From a comparison of divergent path distances from one presubicular class to its major targets, along with a comparison of convergent path distances from each presubicular class to collectively major targets, we found that path distances to the same targets were significantly different between classes, as were the path distances to distinct targets within most classes. This might imply that electrical

impulses reach different targets with varying delays, both within the same class and between classes.

Topographic analysis of presubicular classes revealed spatial separation between the somata of each class. Grid cells are co-localized with head-direction and border cells in dorsal presubiculum as compared to the ventral presubiculum[30], in a manner similar to that found in the deeper layers of the medial entorhinal cortex[31], implying that grid cells are more likely to be found in class A38 and E6 neurons than in class B27 neurons. Topographic analysis also suggests the possibility of anatomically mapping the input and output of the circuitry specializing in head direction computations[32]. Our reported topography of presubicular projections classes is consistent with the recently observed local modularity of the head-direction microcircuit[33], and may help clarify the relationship between the egocentric and allocentric spatial and episodic representations of the cortico-hippocampal system[34]. Previous studies found head-direction cells in

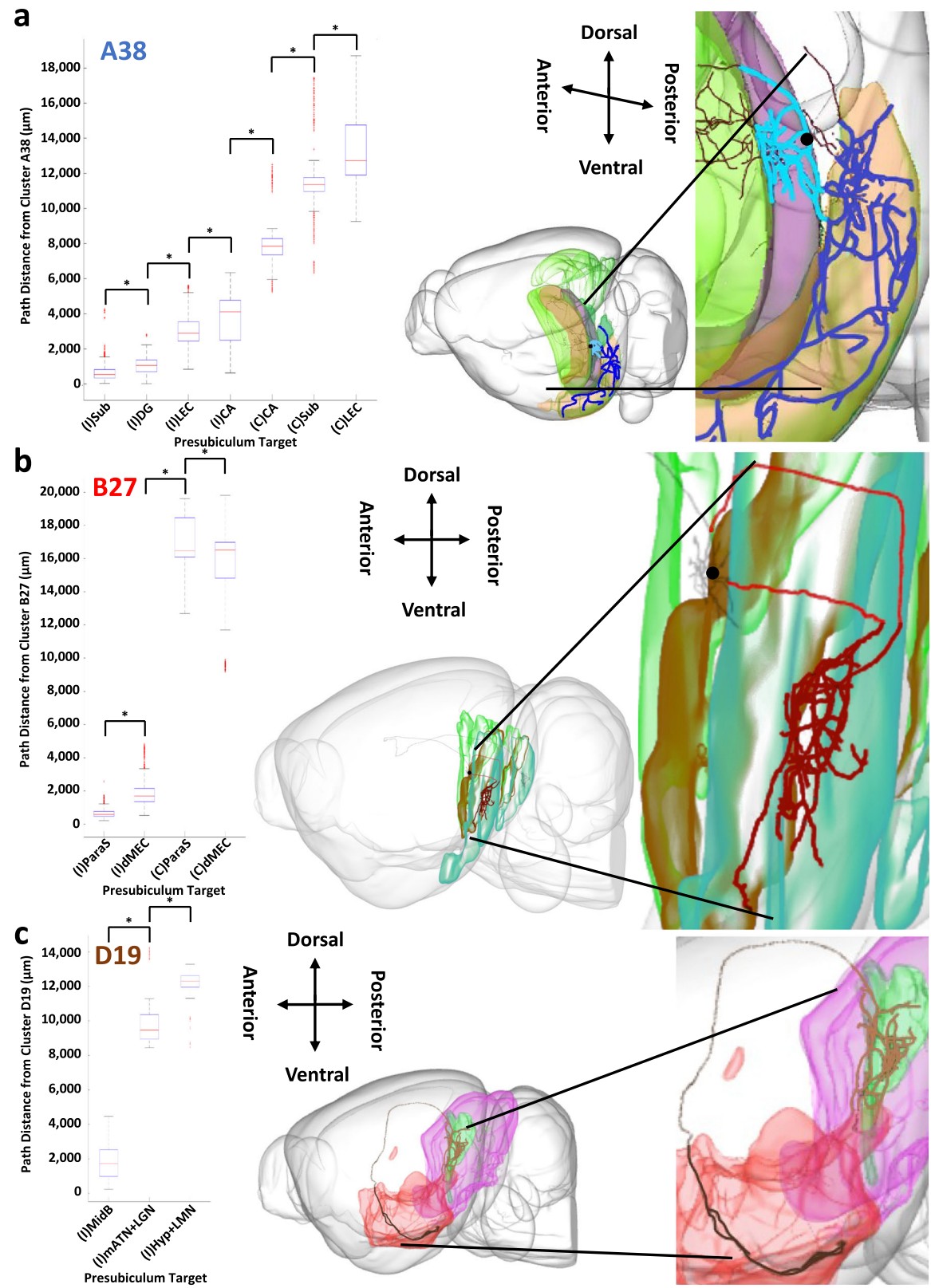

layer 3 of dorsal presubiculum[33]. Since class D19 neurons are found in layer 3, whereas class A38 neurons are mostly confined to layer 2, this would imply that head-direction cells make up part of the composition of class D19, but less so for class A38.

As with many secondary data analyses, we have limited knowledge of, and control over, artifactual shortcomings in the utilized datasets due to possible idiosyncrasies in labeling, imaging, tracing,

registration, and mapping. However, the technique introduced with this work is applicable to many disparate sources of data besides MouseLight, including fMOST[13–15] and even MapSeq/BarSeq[35,36]. These data sources follow separate experimental and computational protocols, allowing independent validation for the source regions in which these datasets overlap. Our results so far, in the cases of the mouse primary motor cortex and presubiculum,

**Fig. 6 | Divergent path distance comparison from one neuron class in the presubiculum to its targets. a** Box and whisker plot depicting the range of the path distances from class A38 to its major ipsilateral (I) and contralateral (C) targets ((I)Sub: $n = 2865$ independent axonal path lengths; (I)DG: $n = 771$; (I)LEC: $n = 18,720$; (I)CA: $n = 957$; (C)CA: $n = 304$; (C)LEC: $n = 17,142$; (C)Sub: $n = 273$). Based on a CCF-registered reconstruction, the axonal path distance of an archetype neuron (AA0159) from class A38 (light blue), from its soma (black) in the ipsilateral pre-subiculum (green) to the subiculum (purple), is significantly shorter than that (dark blue) to the lateral entorhinal cortex (orange). **b** Box and whisker plot depicting the distributions of path distances from class B27 to its major ipsilateral and contralateral targets ((I)ParaS: $n = 1622$ independent axonal path lengths; (I)dMEC: $n = 25,005$; (C)dMEC: $n = 15,800$; (C)ParaS: $n = 1666$). Based on a CCF-registered reconstruction, the axonal path distance of an archetype neuron (AA0374) from class B27 (light red), from its soma (black) in the ipsilateral presubiculum (green) to the parasubiculum (brown), is significantly shorter than that (dark red) to the medial entorhinal cortex, dorsal zone (cyan). **c** Box and whisker plot depicting the path distances from class D19 to its major ipsilateral targets ((I)MidB:

$n = 405$ independent axonal path lengths; (I)mATN+LGN: $n = 349$; (I)Hyp+LMN: $n = 469$). Based on a CCF-registered reconstruction, the axonal path distance of an archetype neuron (AA0031) from class D19 (light brown), from its soma (black) in the ipsilateral presubiculum (green) to the midbrain (magenta), is significantly shorter than that (dark brown) to the hypothalamus and lateral mammillary nucleus (red). See Fig. 4 for abbreviation definitions. The red horizontal lines in the box and whisker plots depict the medians. The first quartiles (Q1) and the third quartiles (Q3) are represented, respectively, by the lower and upper bounds of the boxes. Error bars represent the data range, where the lower line is $Q1 - 1.5 \times (Q3 - Q1)$ and the upper line is $Q3 + 1.5 \times (Q3 - Q1)$. Red pluses are outlier data points that are greater than $Q3 + 1.5 \times (Q3 - Q1)$ or less than $Q1 - 1.5 \times (Q3 - Q1)$. In all depicted comparisons, significant differences in distances were calculated using a two-sided Wilcoxon Signed Rank Test performed on neuronal path distances and multiple testing was corrected for by False Discovery Rate to determine the significance of the resultant $p$-values. A * indicates that the path differences were found to be significant. Source data are provided as a Source Data file. The neurons and brain depictions in all panels are from the Janelia MouseLight project.

indicate that the executed analysis is robust to these possible confounding variables[22].

Overall, this study revealed that neurons can be divided into distinct classes based on axonal projection patterns, as demonstrated in layer 6 of the primary motor cortex and the presubiculum. Our applied analyses can be used to similarly analyze neurons projecting from all other mouse brain regions with sufficient data. There are currently approximately 40 regions fitting this criterion in the existing datasets, but this number is expected to grow in the near future. Furthermore, we suggest the application of pairing Levene's test and unsupervised hierarchical clustering to other complementary datasets, such as single-cell transcriptomic datasets, to classify neurons across a molecular domain, in addition to an anatomical domain, as demonstrated here. Moreover, all these complementary datasets are broadly expected to continue to grow in sample size, brain coverage, and acquisition pace[37,38], supporting a call to establish cloud-based, community accessible pipelines for robust, rigorous, and systematic neuronal characterization[39,40].

## Methods

### Choice of metric to quantify axonal extent

The axonal reconstructions utilized in this study are represented in the Janelia MouseLight public repository[21] (available at http://ml-neuronbrowser.janelia.org) as SWC-formatted files[41]. This standard data structure captures each neuronal tracing point with a set of numerical values that include the three-dimensional coordinates, the local neurite radius, and the identity of the next point in the path to the root[42]. The spacing between consecutive points can be computed as 3D Euclidean distance of their locations, and the length of the axon as the sum of those distances.

It may be tempting to assume that length constitutes the most natural metric to quantify axonal extent in each brain region. However, it is important to remember that this dataset was collected by light microscopy and does not capture the distribution of presynaptic boutons. Therefore, it is not directly possible to distinguish synapse-bearing portions of the axonal arborization from fibers of passage. It is arguably the connectivity target regions that should guide classification rather than the regions through which the projection simply travels to reach its destinations. This can be a critical confounding factor as the longest unbranching stretches of cortical projecting neurons often correspond precisely to fibers of passage[43].

In our own axonal reconstruction experience, we noticed that, while tracking branches from the image stack, it is natural to increase the density of tracing points when the arbor meanders in the synaptic neuropil than when it traverses layers devoid of potential postsynaptic

partners[10]. Moreover, when we painstakingly identified and annotated the position of all axonal boutons in a different study, we found a tendency to utilize more tracing points per unit of length in bouton-rich branches than otherwise[44]. These observations are consistent with the need for greater sampling rates in the presence of larger signal gradients or first derivatives in terms of axonal curvature (neuropil meandering), radius (bouton swelling vs. shaft), or both (bifurcation points).

In the MouseLight dataset analyzed here (Source data are provided as a Source Data file), the number of points in an axonal branch and the corresponding branch length are significantly linearly correlated (Pearson $R = 0.742$; $N = 19,847$; $p < 10^{-99}$). To determine whether the average spacing between points varies non-uniformly between supposed fibers of passage and putative synapse-bearing axons, we separated the axonal branches in each presubiculum neuron based on Strahler (centripetal) order, namely order 1–3 (terminal, pre-terminal, and pre-pre-terminal branches) from order 4–6 (those more than 2 bifurcations away from an ending). This choice is justified by converging experimental evidence that cortical axons make most presynaptic contacts at Strahler orders 1–3, while boutons are substantially sparser at orders 4–6[45,46]. This is also consistent with the strongly non-uniform distribution of average branch length in the dataset investigated in this study, indicating more likely fibers of passage at Strahler order 4–6 ($964.4 \pm 1037.1\ \mu m$) than at Strahler order 1–3 ($144.7 \pm 80.2\ \mu m$; one-tail $t$-test $p = 4.8 \times 10^{-11}$; $t$-value $= -7.35$; df $= 88$). We found indeed that the average spacing of tracing points is significantly smaller at order 1–3 ($22.54 \pm 10.61\ \mu m$) than at Strahler order 4–6 ($39.07 \pm 27.43\ \mu m$; one-tail $t$-test $p = 3.5 \times 10^{-7}$; $t$-value $= -5.26$; df $= 111$). This again supports the notion that the number of tracing points is a better proxy indicator of synapse-bearing axonal extent than total length. We thus chose to utilize the number of tracing points, and not arbor length, as the metric to classify axonal projections.

### Data extraction and storage

The location of each axonal data point for nearly 1100 neurons was extracted from JSON files from the MouseLight dataset[21] using the freeware JSONLab v1.5 (v2.0 is now available at https://sourceforge.net/projects/iso2mesh/files/jsonlab/2.0%20%28Magnus%20Prime%29/jsonlab-2.0.zip/download), where the three-dimensional coordinates and parcel information were provided for each axonal point of the neuron. The number of axonal points in each brain parcel were tabulated for all neurons and were stored in a matrix, in which each row represents a neuron, each column represents a parcel, and the values in each cell represent the axonal counts of a particular neuron in a particular region (Fig. 1; Source data are provided as a Source Data file).

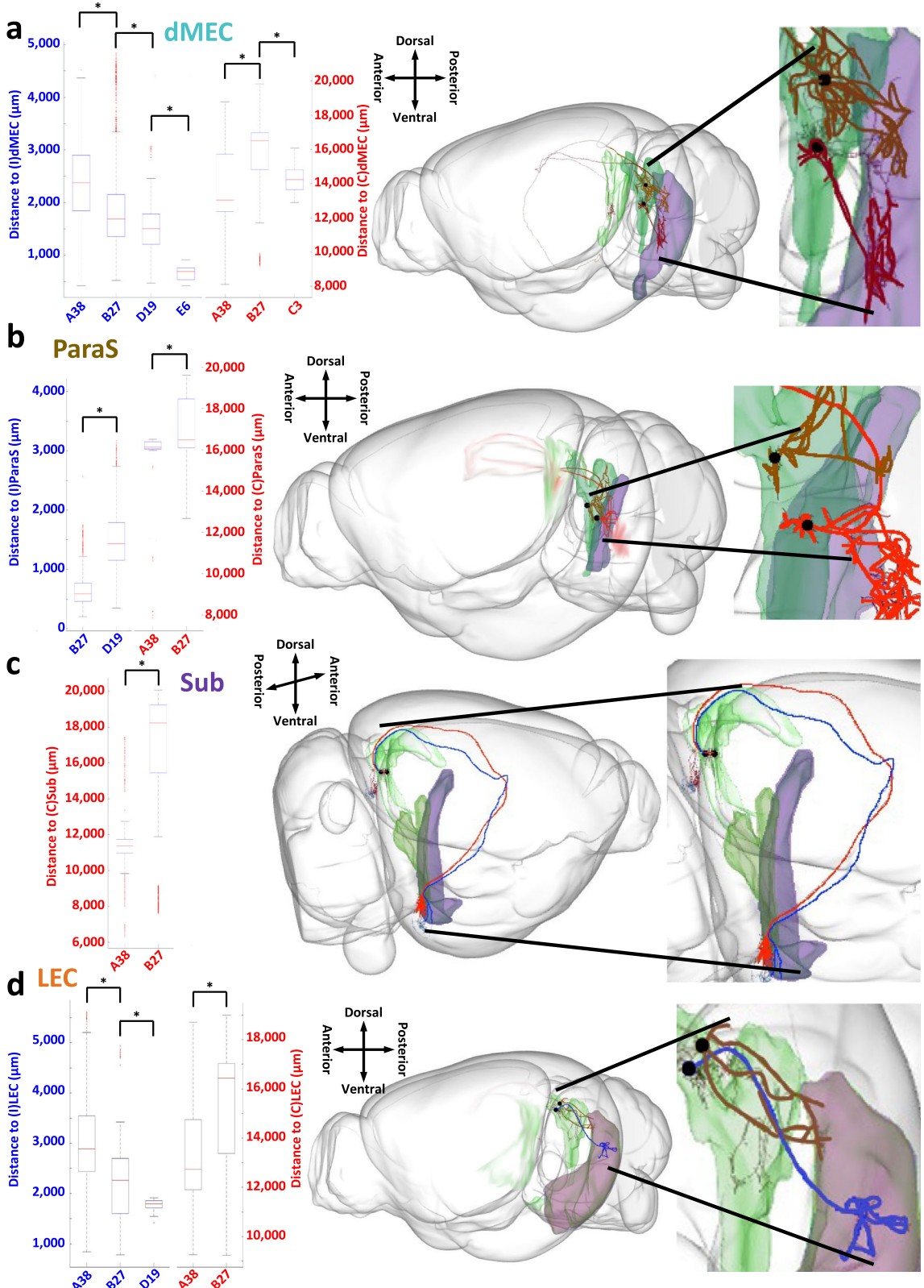

## Hypothesis design

To determine whether distinct projection classes of neurons exist from a particular parcel of the brain, hypothesis $H_A$, we tested the pairwise differences between neurons from the experimental matrices described above. If only a single class of neurons exists, then only a single distribution of differences between neurons will be generated (Fig. 2a). If two hypothetical classes exist, then the differences between neurons, evaluated two at a time, will be smaller within a given class than across the two classes (Fig. 2b, c). In a multi-class scenario, a histogram of the differences between neurons should be wider than the distribution generated when all the neurons belong to just a single class (Fig. 2d). To generate the distribution of differences for the null hypothesis, $H_0$, a randomized control matrix was generated from the original experimental matrix through multiple iterations of the

**Fig. 7 | Convergent path distance comparison from each presubiculum cluster to major targets. a** Box and whisker plot depicting the range of the path distances from neurons in the various classes to the ipsilateral (I) and contralateral (C) medial entorhinal cortex, dorsal zone (A38 to (I)dMEC: *n* = 6194 independent axonal path lengths; B27 to (I)dMEC: *n* = 25,005; D19 to (I)dMEC: *n* = 1030; E6 to (I)dMEC: *n* = 18; A38 to (C)dMEC: *n* = 4301; B27 to (C)dMEC: *n* = 15,800; C3 to (C)dMEC: *n* = 183). Based on CCF-registered reconstructions, the axonal distance of an archetype neuron from class B27 (red, AA0526), from its soma in the presubiculum (green) to the ipsilateral dMEC (purple), is significantly longer than the comparable distance of an archetype neuron from class D19 (brown, AA0875). **b** Box and whisker plot depicting the path distances from neurons in various classes to the ipsilateral and contralateral parasubiculum (B27 to (I)ParaS: *n* = 1622 independent axonal path lengths; D19 to (I)ParaS: *n* = 2031; A38 to (C)ParaS: *n* = 76; B27 to (C)ParaS: *n* = 1666). Based on CCF-registered reconstructions, the axonal distance of an archetype neuron from class B27 (red, AA0377), from its soma in the presubiculum (green) to the ipsilateral ParaS (purple), is significantly shorter than the comparable distance of an archetype neuron from class D19 (brown, AA0385). **c** Box and whisker plot of the path distances from neurons in various classes to the contralateral subiculum (A38 to (C)Sub: *n* = 273 independent axonal path lengths; B27 to (C)Sub: *n* = 1389). Based on CCF-registered reconstructions, the axonal distance of an archetype neuron from class A38 (blue, AA0528), from its soma in the presubiculum (green)

to the contralateral Sub (purple), is significantly shorter than the comparable distance of an archetype neuron from class B27 (red, AA0526). **d** Box and whisker plot depicting the path distances from neurons in various classes to the ipsilateral and contralateral lateral entorhinal cortex (A38 to (I)LEC: *n* = 18,720 independent axonal path lengths; B27 to (I)LEC: *n* = 2002; D19 to (I)LEC: *n* = 27; A38 to (C)LEC: *n* = 17,142; B27 to (C)LEC: *n* = 4532). Based on CCF-registered reconstructions, the axonal distance of an archetype neuron from class D19 (brown, AA0912), from its soma in the presubiculum (green) to the ipsilateral LEC (purple), is significantly shorter than the comparable distance of an archetype neuron from class A38 (blue, AA0878). See Fig. 4 for abbreviation definitions. The red horizontal lines in the box and whisker plots depict the medians. The first quartiles (Q1) and the third quartiles (Q3) are represented, respectively, by the lower and upper bounds of the boxes. Error bars represent the data range, where the lower line is Q1 − 1.5 × (Q3 − Q1) and the upper line is Q3 + 1.5 × (Q3 − Q1). Red pluses are outlier data points that are greater than Q3 + 1.5 × (Q3 − Q1) or less than Q1 − 1.5 × (Q3 − Q1). In all depicted comparisons, significant differences in distances were calculated using a two-sided Wilcoxon Signed Rank Test performed on neuronal path distances and multiple testing was corrected for by False Discovery Rate to determine the significance of the resultant *p*-values. A * indicates that the path differences were found to be significant. Source data are provided as a Source Data file. The neurons and brain depictions in all panels are from the Janelia MouseLight project.

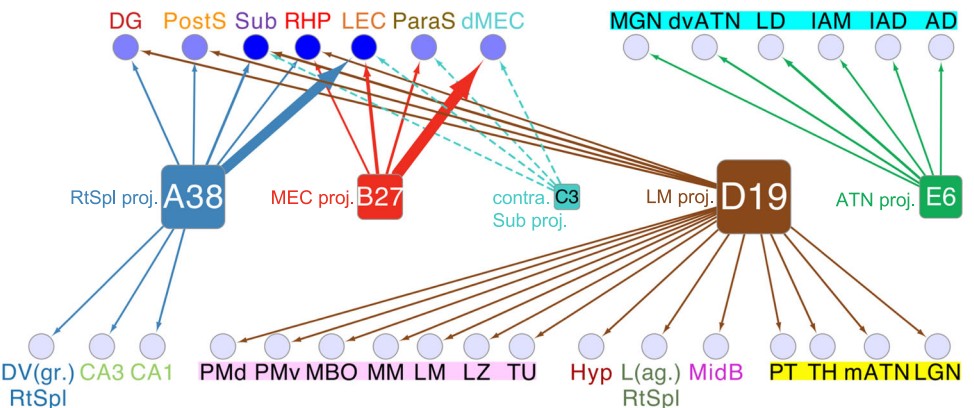

**Fig. 8 | Summary diagram of presubicular class connectivity.** The diagram summarizes the divergence of projections from the classes of the presubiculum and the convergence into parcels distributed throughout the brain. The sizes of the class nodes are proportional to the population sizes of the given classes. The correspondence to prior classification of presubicular convergence targets are listed to the left of the class nodes. The thickness of the arrows is proportional to the

number of axonal points in the destination parcel. The dashed arrows represent contralateral connections. The intensity of blue of the destination parcel nodes corresponds to the number of converging connections, where darker corresponds to more connections. From within each cluster, the arrow lengths are ranked according to the path distance to target. See Fig. 4 for the parcel abbreviation definitions.

stochastic pairwise swapping of axonal counts from two neurons across two target regions (Fig. 2e). This method randomized the projection patterns, yielding a continuum consistent with the regional connectivity of Fig. 2a, while preserving axonal sizes (row sums) and regional targeting (column sums) of the original experimental matrix.

## Levene's test
We assessed the hypothesis that the variance of experimental data was significantly larger than the variance of randomized data (α = 0.05). For both the experimental and randomized matrices, we computed the arccosine between a pair of neuronal vectors, each composed of the axonal counts across all target regions (https://github.com/Projectomics/MATLAB). These angles measure the projection difference of two neurons across all brain parcels. We then performed a 1-tailed Levene's test[47] on the angle distributions of the experimental and randomized matrices to assess whether their variances differed significantly. To this aim, we used the MATLAB function vartestn with the TestType parameter set to LeveneAbsolute. If the experimental data had a greater variance than the randomized data, then the experimental data could be further divided into classes, consistent with the scenario presented in Fig. 2b.

## Unsupervised hierarchical clustering
We used unsupervised agglomerative hierarchical clustering to determine a biologically accurate division of neuron classes based on axonal projection patterns. Specifically, the MATLAB linkage function, with the average algorithm for computing distance between clusters, was utilized on the 93 MouseLight neurons originating in the presubiculum and the 52 MouseLight neurons originating in layer 6 of the primary motor cortex. The initial assumption (null hypothesis) was that all neurons were part of a single class. If Levene's test yielded significant results, the number of class divisions was incremented, and the technique was again repeated on each class division. This iterative process continued until none of the subdivided classes yielded significant results, thereby yielding the final class divisions (Fig. 2f).

## Non-negative least squares
To estimate the fractional counts of cells in each of k projection classes in each region, we matched their respective single-cell axonal patterns against the regional connectivity from anterograde tracing to the m known targets, as presented in the Allen Mouse Brain Connectivity Atlas (http://connectivity.brain-map.org/projection). The problem is

equivalent to a set of constrained, weighted, linear equations that can be solved numerically by non-negative least-square (NNLS) optimization[48]. NNLS finds the values **x** that minimizes the Euclidean norm of (**Ax** - **b**) with the constraint $x \geq 0$[49], where **x** is the k-dimensional vector representing the fractions of neurons in each class; **b** is the m-dimensional vector representing the weights of the regional projections to each target; and **A** is a k-by-m matrix with rows representing the projections of each class (the sum of the summary vectors of the corresponding neurons) and columns representing target regions. NNLS was computed using the lsqnonneg function in MATLAB.

Matrix **A** and vector **b** were based on data from the MouseLight dataset (Source data are provided as a Source Data file) and the Allen Mouse Brain Connectivity Atlas, respectively. Setting the target region to the whole brain in the Connectivity Atlas and the source region to the presubiculum resulted in 7 tracing experiments, which included projection volumes and projection densities for each target brain region. Cross referencing the targeted regions of the MouseLight axonal projections with target regions that appeared in all 7 anterograde tracing experiments resulted in a listing of 66 regions. Matrix **A** was created with rows representing these 66 brain regions and columns representing the 5 neuron classes found by pairing Levene's test with unsupervised hierarchical clustering of the presubiculum data (Source data are provided as a Source Data file). The average projection volume and density values for each of the 66 regions were calculated from the 7 experiments, and the averages were multiplied to populate the columns of vector **b**.

To obtain the highest confidence in the NNLS analysis, matrix **A** was sequentially bi-normalized first by axonal length and then by invaded region (Source data are provided as a Source Data file). Specifically, first each cell in matrix **A** was normalized so that each row summed to one. Next, each value was divided by the number of regions, 66, and multiplied by the number of clusters, 5, such that the sum of all values in matrix **A** equaled 5. Subsequently, each cell in matrix **A** was normalized so that each column summed to one. Vector **b** was normalized such that the sum of all values equaled to one. Finally, the squared Euclidean norm of the residual of the MATLAB function lsqnonneg was calculated as a proxy for the uncertainty of the analysis.

### Soma analysis
To quantify the spatial separation among the somata among the neuron projection classes in the presubiculum, we performed a convex hull analysis for the location of the soma centers in each class using MATLAB. To create the convex hull, outliers were removed by iteratively going through all points in each class and calculating the volume of the convex hull without each point. If the volume differed by more than 1/n of the volume of the original convex hull, which included all points, the point was considered an outlier and removed from the dataset. This established an algorithmic thresholding that corresponded well with the visual inspection of potential outliers. However, if removing the outliers resulted in fewer than four somata, the minimal number of points required to conduct a convex hull analysis, all points were considered. Between each pair of convex hulls, the proportion of the volume of overlap to the volume of the union of the convex hulls was used to assess the similarity between topographic locations.

### Analysis of divergence and convergence
Utilizing the original JSON data files, for every neuron in each presubiculum class, we measured the path distance from the soma to each axonal point in the target region. We then calculated the median path distance to each target region across all neurons in the class, and performed a Wilcoxon Signed Rank Test[50], using the MATLAB function ranksum, to assess whether the path distances to each characteristic target of a particular class were significantly different. Using the same data files, we also performed a Wilcoxon Signed Rank Test to assess

whether the path distances to each characteristic target between all clusters were significantly different. In both sets of comparisons, multiple testing was corrected for by False Discovery Rate to determine the significance of the resultant $p$-values.

### Reporting summary
Further information on research design is available in the Nature Portfolio Reporting Summary linked to this article.

## Data availability
Source Data for Figs. 1, 3b, 4b, 6, and 7 are provided with the paper, as are the neurons depicted in Figs. 3d and 4e, data for the Strahler order analysis, the axonal counts for the layer 6 of the primary motor cortex, and the non-negative least squares analysis The Janelia MouseLight reconstructions from layer 6 of the primary motor cortex and the presubiculum are available for citation by way of their individual digital object identifiers, as provided in the Source data Supplement file.

## Code availability
All code is available in the GitHub repository at https://github.com/Projectomics.

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

## Acknowledgements

We thank Dr. Rodrigo Muñoz-Castañeda for help with validating the mapping of neuronal reconstructions to anatomical coordinates. This work was supported in part by NIH grants R01NS39600, U01MH114829, and RF1MH128693, all to G.A.A.

## Author contributions

D.W.W., S.B., S.S., and S.V. contributed to the analysis and interpretation of data, to the writing of software, and to the writing of the manuscript. G.A.A. contributed to the conception of the project, to the analysis and interpretation of data, and to the writing of the manuscript.

## Competing interests

The authors declare no competing interests.
