## [Peer Review File · Nature Communications]

Unsupervised classification of brain-wide axons reveals the presubiculum neuronal projection blueprintREVIEWER COMMENTS

Reviewer #1 (Remarks to the Author):

Review of: Wheeler, DW et al. Unsupervised classification of brain-wide axons reveals neuronal projection blueprint

This study analyzes single neuron axon tracing data from the mouse Light project to classify subtypes of projection neurons of the presubiculum based on their targets. The more than thousand single neuron axonal tracings provided by the MouseLight and other similar projects are quite impressive in demonstrating the diversity of cortical projections. The analysis in the study by Wheeler et al. demonstrates how such data may be used to provide new insights into the functional organization of brain circuit network connectivity.

Comments:

- 1) The main findings from this study are the identification of 5 subtypes of projection neurons of the presubiculum. The title of the paper should include the “presubiculum cortex”, otherwise the title suggests that the study will describe a general “neuronal projection blueprint”
- 2) The 5 subtypes of presubiculum identified appear to correspond directly to 5 known subtypes already identified. In this study, the subtypes are “named” based on the cluster diagram. Would be helpful to identify which of the 5 prior subtypes correspond to A38, B27, C3, D19, E6
- 3) The diagram in fig 4 C has the most useful depiction of the 5 subtypes and their projections. However, the main “new” information about these subtypes is difficult to get from the diagram as it is overly compressed. Expanding this diagram to highlight the comparison of connections (divergence, convergence, subcortical projection patterns) would help.
- 4) I was unable to extract data from the MouseLight public dataset using the link provided: (<https://www.mathworks.com/matlabcentral/ml-downloads/downloads/submissions/33381/versions/22/download/zip>). The message was that the site was unavailable, will try again. Does that link allow downloading the actual JSON file?
- 5) The data for axonal projections appear to be individual points. What is the spacing between points, is it uniform, in other words what is the relationship between the number of points and the actual length of the axon.
- 6) The mouse light axon data does not distinguish terminals from axons of passage. At least some comment should be provided as to how this affects analysis.

Reviewer #2 (Remarks to the Author):

The article proposes a new technique to identify neuronal classes based on axonal projection patterns through coupling rigorous statistical testing with unsupervised hierarchical clustering, entitled “Unsupervised classification of brain-wide axons reveals neuronal projection blueprint”. In particular, they apply Levene’s one-tail statistical test to determine whether such distinct projection classes exist from a source region, and the unsupervised hierarchical clustering determines a division of neuron classes. The method is firstly verified in 52 MouseLight layer 6 neurons from the primary motor cortex, revealing a consistent result with the previous studies in the mammalian brain. Then they perform the same method in the presubiculum of the mouse brain, reporting some new insights successfully. Except for anatomical insights, they also discuss functional domains like spatial representation, sensory input, and auditory and visual pathways.

The data analysis and interpretation are comprehensive, and the paper is quite well written. I think this technique is valuable to be followed and further verified. Several suggestions and questions I have may help improve the paper's overall quality.

(1) The authors suggest the application of pairing Levene’s test and unsupervised hierarchical clustering in the current scenario. Are the other combinations of statistical and clustering methods valid in a similar scenario?

(2) In Figure 4, five clusters are produced in the current scenario. How to definite the number of clusters from the final results? It seems varied along with different partitioning scales.

(3) The proposed method puts focus on neuron classification and gives detailed comparisons from different perspectives. However, the authors seem to give little explanation on path generation. I think that adding more details on plotted neuronal projection path will be friendlier for reads, like how to update traces during the iterative process. It can also help readers understand how to calculate the path distance in Figures 6 and 7.

(4) Figure 4E, clarify the boundary of the figure.

Reviewer #3 (Remarks to the Author):

The manuscript establishes a new method combining statistical testing (Levene's test) with unsupervised hierarchical clustering to identify different neuronal classes or clusters based on their axonal projection patterns. The authors have also validated this approach using MouseLight data on axonal arborisation and neuronal clusters of layer 6 neurons of the mouse primary motor cortex. The authors then apply this novel unsupervised hierarchical clustering method to the presubiculum, a less explored yet crucial area, to identify presubicular neuronal clusters based on their projection patterns, spatial distribution of somata, convergence and divergence of path distances to different target brain regions by neuronal classes of the presubiculum. The introduced unsupervised hierarchical clustering appears robust, the inferences valid and provides novel information of neuronal clusters based on their projection patterns and the factors that determine this, in the presubiculum.

The authors could do with some more elaboration of the details in the figure legends for better understanding of the deductions made. I have included some comments with this regard against the figure legends in the manuscript.

Few other points that need clarification:

1. Lines 94-97 "If the answer is positive, we must discard H_0 and accept H_A . Starting from the top node in an unsupervised hierarchical clustering tree, we can thus repeat Levene's test on the neurons of each of the two subtrees, continuing the process until none of the variance differences are statistically significant."

Perhaps include a line or two here that states explicitly what can be inferred about the clusters if the variances are not statistically significant. This makes it easy for the non-expert reader.

2. Lines 264-268 "This suggests the possibility of anatomically mapping the input and output of the circuitry specializing in head direction computations³⁰. Our reported topography of presubicular projections classes is consistent with the recently observed local modularity of the head-direction microcircuit³¹, and may help clarify the relationship between the egocentric and allocentric spatial and episodic representations of the cortico-hippocampal system³²."

While these interpretations could be valid, a little more explanation on the correlation of spatially separated somata between different classes and head-direction computations will be useful.

3. Other minor amends are annotated as comments in the manuscript.

REVIEWER COMMENTS

Reviewer #1 (Remarks to the Author):

Review of: Wheeler, DW et al. Unsupervised classification of brain-wide axons reveals neuronal projection blueprint

This study analyzes single neuron axon tracing data from the mouse Light project to classify subtypes of projection neurons of the presubiculum based on their targets. The more than thousand single neuron axonal tracings provided by the MouseLight and other similar projects are quite impressive in demonstrating the diversity of cortical projections. The analysis in the study by Wheeler et al. demonstrates how such data may be used to provide new insights into the functional organization of brain circuit network connectivity.

We are grateful to the Reviewer for the in-depth feedback on our paper, the positive evaluation, and the constructive criticism, which we carefully addressed in our revision as explained in detail below.

Comments:

1) The main findings from this study are the identification of 5 subtypes of projection neurons of the presubiculum. The title of the paper should include the “presubiculum cortex”, otherwise the title suggests that the study will describe a general “neuronal projection blueprint”

We thank the Reviewer for the suggestion. We have now re-titled the manuscript to “Unsupervised classification of brain-wide axons reveals neuronal projection blueprint: an illustrative application to the presubiculum.”

2) The 5 subtypes of presubiculum identified appear to correspond directly to 5 known subtypes already identified. In this study, the subtypes are “named” based on the cluster diagram. Would be helpful to identify which of the 5 prior subtypes correspond to A38,B27, C3, D19, E6

We would like to thank the Reviewer for the suggestion. We have added text describing the correspondence between the 5 prior classes based on their main convergence targets and our 5 classes, and we have added appropriate labels to our new summary Figure 8.

3) The diagram in fig 4 C has the most useful depiction of the 5 subtypes and their projections. However, the main “new” information about these subtypes is difficult to get from the diagram as it is overly compressed. Expanding this diagram to highlight the comparison of connections (divergence, convergence, subcortical projection patterns) would help.

We thank the Reviewer for the helpful suggestion. We have now added a Figure 8, and accompanying text, to provide a clear summary of the information presented in the manuscript concerning presubicular projections, including divergence and convergence information, population size, and detailed subcortical patterns.

4) I was unable to extract data from the MouseLight public dataset using the link provided: (<https://www.mathworks.com/matlabcentral/ml-downloads/downloads/submissions/33381/versions/22/download/zip>). The message was that the site

was unavailable, will try again. Does that link allow downloading the actual JSON file?

We apologize for any confusion. We have added text to clarify that the MouseLight dataset is available from a separate repository (link now provided) that is distinct from the software used to extract the data from the JSON files. We have also updated the software link, as its author has updated the software from v1.5, which is used in this manuscript, to v2.0.

5) The data for axonal projections appear to be individual points. What is the spacing between points, is it uniform, in other words what is the relationship between the number of points and the actual length of the axon.

Excellent questions. Please see answers in the reply to the following point (#6) below.

6) The mouse light axon data does not distinguish terminals from axons of passage. At least some comment should be provided as to how this affects analysis.

Thank you for the relevant observation and suggestion. In the MouseLight dataset analyzed here, the number of points in an axonal branch and the corresponding branch length are significantly correlated ($R=0.742$; $N=19,847$; $p<10^{-99}$). To determine whether the average spacing between points is uniform in different portions of the arbor, we analyzed this quantity by Strahler (centripetal) order, which is known to separate synapse-bearing axons from fibers of passage (doi.org/10.1371/journal.pcbi.1000711; doi.org/10.1101/2023.08.07.552361; see also representative image below). **This analysis, now included in the new section entitled “Choice of metric to quantify axonal extent,” supports our choice of using individual points as opposed to axonal length as the classification metric.** Specifically, we found that axonal branches near the terminals (Strahler order 1-3) are between 6- and 7-fold shorter than the axonal branches more than 2 bifurcation away from the terminals (Strahler order 4-6); and that the average spacing between points is approximately half in the axonal branches near the terminals than in the branches farther from terminals.

Reviewer #2 (Remarks to the Author):

The article proposes a new technique to identify neuronal classes based on axonal projection patterns through coupling rigorous statistical testing with unsupervised hierarchical clustering, entitled “Unsupervised classification of brain-wide axons reveals neuronal projection blueprint”. In particular, they apply Levene’s one-tail statistical test to determine whether such distinct projection classes exist

from a source region, and the unsupervised hierarchical clustering determines a division of neuron classes. The method is firstly verified in 52 MouseLight layer 6 neurons from the primary motor cortex, revealing a consistent result with the previous studies in the mammalian brain. Then they perform the same method in the presubiculum of the mouse brain, reporting some new insights successfully. Except for anatomical insights, they also discuss functional domains like spatial representation, sensory input, and auditory and visual pathways.

The data analysis and interpretation are comprehensive, and the paper is quite well written. I think this technique is valuable to be followed and further verified. Several suggestions and questions I have may help improve the paper's overall quality.

We are grateful to the Reviewer for the in-depth feedback on our paper, the positive evaluation, and the constructive criticism, which we carefully addressed in our revision as explained in detail below.

(1) The authors suggest the application of pairing Levene's test and unsupervised hierarchical clustering in the current scenario. Are the other combinations of statistical and clustering methods valid in a similar scenario?

We thank the Reviewer for the question. We have added a paragraph to the Discussion describing how Levene's test was chosen because it is not dependent on the distributions being normal, which is not guaranteed given the size of the MouseLight data, when separated out into the different originating parcels. Another form of unsupervised clustering could be utilized to achieve similar ends. The important aspect is that the method be unsupervised, such that the data themselves direct the clustering without any user input.

(2) In Figure 4, five clusters are produced in the current scenario. How to definite the number of clusters from the final results? It seems varied along with different partitioning scales.

We apologize for the confusion on our part. We had inadvertently included an extra dashed partition line to Figure 4A, which we have now removed. There are now 5 partition lines corresponding to the 5 clusters. As now more clearly explained in the text, every partition is independently determined by the Levene test without user intervention. Thus, the position of each partition exclusively depends on the statistical significance of the corresponding clusters.

(3) The proposed method puts focus on neuron classification and gives detailed comparisons from different perspectives. However, the authors seem to give little explanation on path generation. I think that adding more details on plotted neuronal projection path will be friendlier for reads, like how to update traces during the iterative process. It can also help readers understand how to calculate the path distance in Figures 6 and 7.

Thank you for the useful suggestion. Actual three-dimensional reconstructions are embedded in a virtual mouse brain in Figures 1, 3c-d, 4d-e, 6, and 7, based on the registration of the MouseLight tracings to the Common Coordinate Framework. In contrast, the classification dendrograms in Figures 3a and 4a are produced by agglomerative hierarchical clustering as now also clearly mentioned in the Methods. We have added text clarifying these facts to the associated figure captions.

(4) Figure 4E, clarify the boundary of the figure.

Thank you for bringing this to our attention. We have now added text clarifying that the parcels highlighted in Figures 4d and 4e are the same as those depicted in Figure 4c, using the same color encoding.

Reviewer #3 (Remarks to the Author):

The manuscript establishes a new method combining statistical testing (Levene's test) with unsupervised hierarchical clustering to identify different neuronal classes or clusters based on their axonal projection patterns. The authors have also validated this approach using MouseLight data on axonal arborisation and neuronal clusters of layer 6 neurons of the mouse primary motor cortex. The authors then apply this novel unsupervised hierarchical clustering method to the presubiculum, a less explored yet crucial area, to identify presubicular neuronal clusters based on their projection patterns, spatial distribution of somata, convergence and divergence of path distances to different target brain regions by neuronal classes of the presubiculum. The introduced unsupervised hierarchical clustering appears robust, the inferences valid and provides novel information of neuronal clusters based on their projection patterns and the factors that determine this, in the presubiculum.

The authors could do with some more elaboration of the details in the figure legends for better understanding of the deductions made. I have included some comments with this regard against the figure legends in the manuscript.

We are grateful to the Reviewer for the in-depth feedback on our paper, the positive evaluation, and the constructive criticism, which we carefully addressed in our revision as explained in detail below.

Few other points that need clarification:

1. Lines 94-97 "If the answer is positive, we must discard H_0 and accept H_A . Starting from the top node in an unsupervised hierarchical clustering tree, we can thus repeat Levene's test on the neurons of each of the two subtrees, continuing the process until none of the variance differences are statistically significant."

Perhaps include a line or two here that states explicitly what can be inferred about the clusters if the variances are not statistically significant. This makes it easy for the non-expert reader.

We thank the Reviewer for the suggestion. We have added text to explain that when Levene's test fails, then, because every cutoff is determined independently of the other points of failure, each cluster is independent of the other clusters. What can be inferred is that all neurons within a cluster (i.e., "under" the same Levene failure point) are statistically equivalent with respect to the axonal projection patterns across the target regions. We also now specify that there is no correspondence between the cutoff levels and the resulting number of neurons in each cluster.

2. Lines 264-268 "This suggests the possibility of anatomically mapping the input and output of the circuitry specializing in head direction computations³⁰. Our reported topography of presubicular projections classes is consistent with the recently observed local modularity of the head-direction microcircuit³¹, and may help clarify the relationship between the egocentric and allocentric spatial and episodic representations of the cortico-hippocampal system³²."

While these interpretations could be valid, a little more explanation on the correlation of spatially separated somata between different classes and head-direction computations will be useful.

We thank the Reviewer for the suggestion. We have added text describing that Balsamo et al. (2022) found head-direction cells in layer 3 of dorsal presubiculum. According to our results summarized in Figure 5b, class D19 neurons are found in layer 3, whereas class A38 neurons are mostly confined to layer 2. This would imply that head-direction cells could make up part of the composition of class D19, but less so for class A38.

3. Other minor amends are annotated as comments in the manuscript.

- a) Lines 148-150: "Class D19, reaching the midbrain, hypothalamus, lateral (agranular) retrosplenial, and the lateral geniculate (visual thalamus) accounted for a plurality (38.1%) of neurons."
meaning a heterogeneous population of neurons? could be more specific

We thank the Reviewer for the question. We utilized the meaning of "plurality" as the set with more elements than any other set, without however reaching an absolute majority. In other words, if one were to more fully sample neurons from across the presubiculum, more neurons would be found to arise from class D19 than from any other class, but still D19 would account for fewer than half of all projection neurons. We have added text to clarify this point and now removed the term 'plurality' to avoid possible confusion.

- b) Lines 573-576: "(F) Unsupervised hierarchical clustering groups a set of neurons into classes based on their relative pairwise differences or similarities, as modeled by a binary dendrogram. The top (root) of the dendrogram represents all neurons lumped into the same class, while the bottom (leaves) shows all neurons split into separate classes."
Add few lines about how the p values were obtained.

We thank the Reviewer for the suggestion. We have added text describing that the p value that determines whether to keep splitting is derived from Levene's test, which is based on a one-way ANOVA of the absolute values of the data values and of the group means to which the data values belong.

- c) Lines 586-587: "The two black dots indicate the cell body locations of the two representative cells from each class."
Can't locate the black dots

We thank the Reviewer for pointing this out. We have now clearly marked the black dots in Figure 3c.

- d) Lines 587-588: "(D) Axonal pathways of all IT and CT neurons in the MouseLight sample (same color coding)."
From the colormap, the thalamic targets seem more numerous. Is this also reflected in C and D?

The axonal points in the thalamic targets are indeed more numerous by a factor of two. This is however difficult to visualize in Figures 3c-d. A breakdown of the axonal counts

making up the colormap in Figure 3b are now provided in the “Source Data.” We have added text to clarify this point.

- e) Line 655: “Left sagittal view of A38 and B27.”
Can some inferences be made from this? Please mention. Same applies to (C) and (D).

We thank the Reviewer for the suggestion. We have added text stating that grid, head-direction, and border cells are found to be plentiful in dorsal presubiculum, implying that class A38 and class E6 neurons are more likely to consist of these cell types than class B27 neurons.

- f) Line 672: “(brown)”
Not clear in the figure.

We have re-created the figure to emphasize the color of the parasubiculum more optimally.

- g) Line 673: “(cyan)”
Not clear in the figure.

We have re-created the figure to better emphasize the color of the dorsal medial entorhinal cortex.

REVIEWERS' COMMENTS

Reviewer #1 (Remarks to the Author):

The revisions the authors have made adequately addressed my comments.

Reviewer #2 (Remarks to the Author):

Thank you for your reply with detailed interpretations. The revised manuscript can help readers grasp the key points more easily. The revised and added figures also meet the requirement of publication.

Reviewer #3 (Remarks to the Author):

The authors have addressed all my concerns. I endorse the publication of the manuscript in its current form.

REVIEWERS' COMMENTS

Reviewer #1 (Remarks to the Author):

The revisions the authors have made adequately addressed my comments.

Reviewer #2 (Remarks to the Author):

Thank you for your reply with detailed interpretations. The revised manuscript can help readers grasp the key points more easily. The revised and added figures also meet the requirement of publication.

Reviewer #3 (Remarks to the Author):

The authors have addressed all my concerns. I endorse the publication of the manuscript in its current form.

There are no reviewers' comments to be addressed at this time.